

# Multi-gene incongruence consistent with hybridisation in *Cladocopium* (Symbiodiniaceae), an ecologically important genus of coral reef symbionts

Joshua I. Brian, Simon K. Davy and Shaun P. Wilkinson

School of Biological Sciences, Victoria University of Wellington, Wellington, New Zealand

## ABSTRACT

Coral reefs rely on their intracellular dinoflagellate symbionts (family Symbiodiniaceae) for nutritional provision in nutrient-poor waters, yet this association is threatened by thermally stressful conditions. Despite this, the evolutionary potential of these symbionts remains poorly characterised. In this study, we tested the potential for divergent Symbiodiniaceae types to sexually reproduce (i.e. hybridise) within *Cladocopium*, the most ecologically prevalent genus in this family. With sequence data from three organelles (*cob* gene, mitochondrion; $psbA^{ncr}$ region, chloroplast; and ITS2 region, nucleus), we utilised the Incongruence Length Difference test, Approximately Unbiased test, tree hybridisation analyses and visual inspection of raw data in stepwise fashion to highlight incongruences between organelles, and thus provide evidence of reticulate evolution. Using this approach, we identified three putative hybrid *Cladocopium* samples among the 158 analysed, at two of the seven sites sampled. These samples were identified as the common *Cladocopium* types C40 or C1 with respect to the mitochondria and chloroplasts, but the rarer types C3z, C3u and C1# with respect to their nuclear identity. These five *Cladocopium* types have previously been confirmed as evolutionarily distinct and were also recovered in non-incongruent samples multiple times, which is strongly suggestive that they sexually reproduced to produce the incongruent samples. A concomitant inspection of next generation sequencing data for these samples suggests that other plausible explanations, such as incomplete lineage sorting or the presence of co-dominance, are much less likely. The approach taken in this study allows incongruences between gene regions to be identified with confidence, and brings new light to the evolutionary potential within Symbiodiniaceae.

# INTRODUCTION

Coral reefs are a highly diverse and important ecosystem, yet are significantly threatened by anthropogenically-driven climate change (*Hughes et al., 2017*). In order for coral reefs to survive the stresses of a changing climate, genetic adaptation over rapid evolutionary timescales has to occur. Adaptation in the coral itself may go some way to provisioning for

Corresponding author
Simon K. Davy,
simon.davy@vuw.ac.nz

the environmentally challenging conditions predicted to come (*Rodriguez et al., 2009*). However, given that the response of corals to environmental conditions is inextricably linked to the diversity and performance of their intracellular symbionts (dinoflagellates of the family Symbiodiniaceae, *LaJeunesse et al., 2018*), increasing attention is being focused on the evolutionary potential within this family.

Coral symbionts have been thought to be exclusively asexual *in hospite* (*Trench, 1997*; *LaJeunesse, 2005*), thanks to their isolated position sequestered inside host cells, and the hypothesis that endosymbiotic sex would encourage exploitation of the host (*Law & Lewis, 1983*). However, previous work in other taxa has shown that intracellular symbionts can sexually reproduce (*Chesnick & Cox, 1987*). In general, it is thought that many such organisms may have cryptic sexual cycles that have previously been unappreciated, in addition to the production of clonal populations via asexual reproduction (*Heitman, 2010*). Now, there is significant evidence that Symbiodiniaceae also displays a mixed reproductive strategy, with periods of asexuality interspersed with occasional to frequent sex (*Thornhill et al., 2017*). While it has never been explicitly observed, there are distinct and observable traces of sex in their genomes (*Baillie et al., 2000*; *LaJeunesse, 2001*; *Santos & Coffroth, 2003*; *Santos et al., 2004*; *Pettay et al., 2011*; *Baums, Devlin-Durante & LaJeunesse, 2014*; *Chi, Parrow & Dunthorn, 2014*; *LaJeunesse et al., 2014*; *Thornhill et al., 2014*; *Levin et al., 2016*). However, these studies have been largely focused on a micro-scale, population level (i.e. intraspecific sex). By contrast, sex between diverse symbiont lineages ('hybridisation') has received little attention in the literature (but see *Wilkinson et al., 2015*). Given the highly thermally stressful conditions predicted by the end of the century (*Kirtman et al., 2013*), the mechanism of hybridisation could potentially have significant and vital adaptive value. By mixing diverse pools of genetic material, hybridisation can allow for rapid adaptation, facilitating macro-evolutionary jumps (*Willis et al., 2006*; *Dittrich-Reed & Fitzpatrick, 2013*). Introgressive hybridisation, where the F1 hybrids subsequently mate with one or both parent populations, can transfer a large quantity of genetic material between the two parent lineages in the space of a few generations. In addition, hybridisation can also produce offspring with elevated fitness ('hybrid vigour'), which can even outcompete the parent species (*Ellstrand & Hoffman, 1990*; *Rhymer & Simberloff, 1996*). Importantly, instances of hybridisation have also been shown to increase in taxing conditions (*Rhymer & Simberloff, 1996*; *Moran & Alexander, 2014*). Therefore, the possibility of hybridisation in coral symbionts raises the potential for adaptation at the required pace and scale for survival.

Research on taxa with similar life-histories suggests that hybridisation is plausible. Hybridisation has previously been reported in a range of dinoflagellate genera, including *Dinophysis, Protoperidinium, Preperidinium* and *Diplopsalis* (*Edvardsen et al., 2003*; *Gribble & Anderson, 2007*; *Hart et al., 2007*). There is also evidence from plant-fungi relationships that endosymbionts can successfully hybridise. In particular, the endophytes *Epichloë* spp. are pathogenic or mutualistic fungi that inhabit a wide range of grasses. Hybridisation appears to be a major mechanism for diversification in this genus, and has been reported to occur inside the grasses *Lolium perenne* (*Schardl et al., 1994*), *Festuca arundinacea* (*Tsai et al., 1994*), *Bromus laevipes* (*Charlton et al., 2014*) and *Poa alsodes*

(*Shymanovich et al., 2017*). In several instances, multiple cases of hybridisation have been recorded, and evidence put forward that those hybrids are fitter than non-hybrids (*Schardl et al., 1994*; *Moon et al., 2004*). While Symbiodiniaceae *in hospite* are generally sequestered inside host cells (*Davy, Allemand & Weis, 2012*), the extensive presence of background symbiont populations inside hosts (*Santos, Taylor & Coffroth, 2001*; *Kemp et al., 2015*), the observation that corals themselves hybridise (*Willis et al., 2006*; *Combosch & Vollmer, 2015*), and the existence of a free-living state (*Coffroth et al., 2006*; *Nitschke, Davy & Ward, 2016*) mean that it is highly possible that at some point diverse symbiont communities may interact, with the possibility for sexual reproduction.

The evolutionary potential of hybridisation has not been targeted within Symbiodiniaceae. However, several indirect observations are suggestive of its occurrence, all within *Cladocopium*, the most prevalent genus. *LaJeunesse et al. (2003)* reported an ITS2 sequence variant they called C1c and treated as an intragenomic variant, as it was only observed in Denaturing Gradient Gel Electrophoresis (DGGE) profiles associated with type C1. However, it was then discovered to be an independent type and called C45 (*LaJeunesse, 2005*). Therefore, the additive DGGE pattern shown in *LaJeunesse et al. (2003)* could have in fact resulted from the hybridisation of C1 and C45. *LaJeunesse (2005)* also defined type C3m using the ITS2 region, which has co-dominant characteristics of both C1 and C3, a pattern attributed to either sexual recombination or homoplasy. A similar scenario was also recorded in symbiont type C3h, an apparent intermediary between C3 and C21 (*LaJeunesse et al., 2004*). This time, the pattern was hypothesised to be due to incomplete lineage sorting (ILS) or sexual recombination between the two different types. Indeed, given the unambiguous existence of 'pure' C3 and C21 in the samples, sexual recombination is a credible explanation. Finally, *Wilkinson et al. (2015)* reported two symbiont types but three distinct symbiont populations inside a single *Pocillopora* colony: C100 symbionts, C109 symbionts and symbionts having co-dominant C100 and C109 repeats in the same cell. Again, the extensive presence of the two 'pure' populations means ILS is a less parsimonious explanation than hybridisation. However, it cannot be completely eliminated as a possibility. In addition, this study took place at Lord Howe Island, the world's southern-most coral reef, and therefore may not be widely applicable across less marginal, low-latitude sites.

Hence, there is a body of indirect evidence for sexual recombination between diverse symbiont types (hybridisation *sensu lato*), and this warrants further study. The current study aimed to gather further defendable evidence as to whether hybridisation occurs in coral symbionts. Because it is very difficult to observe hybridisation directly, it is generally inferred through genetic signals. One of the most common of these is incongruence between gene regions. Because nuclear genes are typically inherited biparentally, while organelle genes are inherited uniparentally, sexual reproduction between different species will result in organelle genes resembling one parent only, while the nuclear genome will have clear traces of both parents (*Rieseberg, Whitton & Linder, 1996*). In extreme cases, repeated backcrosses with a parent type can result in organelle capture, where novel, discordant nuclear-organellar combinations are observed (*Folk, Mandel & Freudenstein, 2017*). Following a hybridisation event, selection can also act to produce incongruence

between gene regions: there may be elevated (or reduced) fitness of certain nuclear-cytoplasmic combinations, or selection pressure may be different for nuclear and cytoplasmic genomes (e.g. a greater selection pressure acting on nuclear genes) (*Rieseberg, Whitton & Linder, 1996*). Therefore, identifying incongruence between gene regions is a common method for assessing potential hybridisation (*Planet, 2006*; *Govindarajulu et al., 2015*), and was utilised in the current study.

The chosen location for this study, Atauro Island and the north coast of Timor, is in the Coral Triangle and therefore widely applicable to other important reef systems. The hypothesis tested was that hybridisation between distinct *Cladocopium* genotypes has occurred at these sites, as evidenced by gene regions in separate organelles (*cob*, mitochondrion; ITS2, nucleus; psbA$^{ncr}$, chloroplast) having experienced different evolutionary histories. Defendable evidence of hybridisation would be a significant step towards understanding the evolution of Symbiodiniaceae and potential coral reef persistence in the future.

## MATERIALS AND METHODS

### Data acquisition

This study represents a novel analysis of the data presented in *Brian, Davy & Wilkinson (2019)*. Briefly, 43 coral genera were sampled from four sites at Atauro Island: Beloi Barrier Reef (BBR); Beloi Harbour (BHB); Beloi Lagoon South (BLS); and BSP (Beloi Saddlepatch) (for a complete list of genera see Table S1 of *Brian, Davy & Wilkinson (2019)*). In addition, three sites were sampled on the northern coast of Timor: Hera West (HEW); Lamsana Inlet East (LIE); and Lamsana Inlet West (LIW). In total, 650 samples were collected from the seven sites. The corals in this study were sampled with the permission of the Ministerio da Agricultura e Pescas (permit number LNC-PC0012.VI.16). Symbiont DNA was extracted using a guanidinium protocol, and amplification via PCR was carried out for the *cob*, ITS2 and psbA$^{ncr}$ regions of the symbiont DNA. For full details of PCR reactions and conditions, see the Supplementary Material. The *cob* amplification utilised either Dinocob1F/Dinocob1R (*Zhang, Bhattacharya & Lin, 2005*) or Cob_f1/Cob_r1 (*Pochon et al., 2012*) primers, while the psbA$^{ncr}$ amplification utilised the primers 7.4-Forw/7.8-Rev (*Moore et al., 2003*). Following purification with MagNA solution (*Rohland & Reich, 2012*), the *cob* and psbA$^{ncr}$ were sequenced in the forward direction with traditional Sanger sequencing (Macrogen Inc., Seoul, South Korea). In contrast, the ITS2 region was amplified via next generation sequencing (NGS). Samples underwent an initial amplification with the primers ITSD (*Pochon et al., 2001*) and ITS2Rev2 (*Stat et al., 2009*), with Illumina adapters attached. Amplicons were purified using MagNA solution, and unique forward/reverse index primers (IDT) were annealed to the ends of each amplification using a second short PCR run of eight cycles. Amplifications were quantified and checked for quality using qPCR (Applied Biosystems StepOne instrument), with the primers ITSD/ITS2Rev2. All samples were pooled, with different volumes of each sample added to achieve an equal concentration (final concentration of pooled library: 4 nM DNA). The pooled library was sequenced on a single lane on the Illumina MiSeq platform by the Centre for Genomics and Proteomics, University of

Auckland, New Zealand. As the incongruence tests utilised (see below) require a single sequence *per* sample, the most dominant ITS2 sequence from the NGS in each sample was extracted (an 'ASV' in *Brian, Davy & Wilkinson (2019)*). While this was necessary for analysis, it could lead to interpretational issues (see Discussion). Only samples that had successful sequences for all three gene regions were chosen, as the tests require exactly the same taxa lists for each tree or partition. Further, only samples that could be placed in an unambiguous alignment were used, which eliminated several samples with highly divergent psbA$^{ncr}$ sequences. This left between 18 and 28 samples *per* site ($\bar{x} = 22.6$), with a total of 158 samples used.

## Incongruence tests

Ideally, a statistical test would be able to test the null hypothesis 'Dataset X and Dataset Y are not incongruent', against an alternate hypothesis 'Dataset X and Dataset Y are incongruent.' A test with this explicit hypothesis does not exist for phylogenetic data, so other tests with slightly different hypotheses have been frequently employed as an approximation. Two of these tests were utilised in this study.

The incongruence length difference (ILD) test (*Farris et al., 1994*) uses the criterion of maximum parsimony, and compares two data partitions (nucleotide alignments) X and Y, of arbitrary length. The null hypothesis is that the defined partitions (X, Y) are no more parsimonious (in terms of making a phylogeny) than random partitions generated from a combination of X and Y, while the alternate hypothesis is that the defined partitions are significantly more parsimonious than random partitions. Functionally, this can be used to test if two datasets have undergone separate evolutionary histories (*Planet, 2006*). The implication is that if X and Y are indeed more parsimonious, they encode contrary evolutionary information that is lost when randomised.

The Shimodaira–Hasegawa (SH) test (*Shimodaira & Hasegawa, 1999*) is an explicit tree-based test using the criterion of maximum likelihood (ML), comparing how well phylogenetic trees explain alignment data. The null hypothesis is that all tested trees are equally good explanations of the data, while the alternate is that some or all tested trees are not equally good explanations of the data. In practice, this test identifies the best tree for a given dataset (i.e. a multiple sequence alignment), and then presents output as to whether other candidate trees are statistically distinct from that best tree. The output hence appears as pairwise comparisons between two trees. This procedure can be used to test for incongruence in datasets X and Y, using trees $T_X$ and $T_Y$ made from those datasets. If $T_X$ and $T_Y$ are equally likely for all or most characters in X and in Y (tested in two separate tests), the test will find a *p*-value > 0.05, and it can be concluded that X and Y are not incongruent, as their trees do an equally good job of explaining each other's data. If they are incongruent, it is expected that $T_X$ will be significantly better than $T_Y$ when considering dataset X, and vice versa for $T_Y$ and Y. The Approximately Unbiased (AU) test was developed by *Shimodaira (2002)* as a derivation of the SH test, and generally finds more accurate results when there are many candidate trees, or some trees are particularly unlikely (*Shimodaira, 2002*; *Strimmer & Rambaut, 2002*); the AU test was hence utilised for testing procedures.

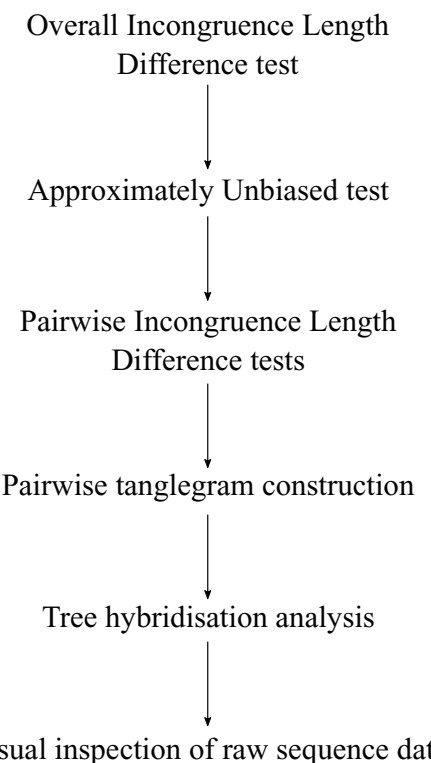

Overall Incongruence Length
Difference test

Approximately Unbiased test

Pairwise Incongruence Length
Difference tests

Pairwise tanglegram construction

Tree hybridisation analysis

Visual inspection of raw sequence data

**Figure 1 Stepwise analyses performed *per* site to identify incongruences in *Cladocopium*.**

To identify incongruence, these two tests (ILD, AU) in addition to other analyses described below were conducted in stepwise fashion (Fig. 1).

## Data assembly

Alignments were created and manually edited in Geneious v8.0.5 (Biomatters, http://www.geneious.com), using the built-in Geneious alignment algorithm with all default settings (gap open penalty = 12, extension = 3). Each site (BBR, BHB, BLS, BSP, HEW, LIE, LIW) had a separate alignment for each gene region (*cob*, psbA$^{ncr}$, ITS2), leading to 21 alignments. Additional holistic datasets for each gene region were created for Atauro Island (92 samples) and Timor (66 samples), to facilitate broad-scale island comparisons. All alignments had 787, 369 and 531 columns for the *cob*, ITS2 and psbA$^{ncr}$ regions, respectively. In total, 27 separate datasets were assembled (three marker regions × (seven sites + two main islands)). Datasets can be accessed online at github.com/brianjosh/Cladocopium_alignments. *Durusdinium glynnii* (D1) was used as the outgroup for these analyses (GenBank Accession Numbers: KY131780 (*cob*); JN558075 (ITS2); MH329571 (psbA$^{ncr}$)). Gaps were coded as a fifth character state. All analyses described below used the program PAUP* 4.0a161 (*Swofford, 2002*) unless otherwise specified. Note that in PAUP*, the ILD test is called the partition homogeneity test.

## Incongruence length difference tests

The three gene regions were concatenated for each site, with each region then treated as a separate partition (*cob*: 1–787; ITS2: 788–1,156; psbA$^{ncr}$ 1,157–1,687; total of 1,687

columns). This was carried out for each site, plus for Atauro Island samples and Timor samples as above (total of nine different concatenations). The individual site analyses were originally carried out with 100 replications, using a MaxTrees value (number of trees stored at any one time) of 1,000. For results that had $p$-values < 0.2, a more thorough confirmatory analysis was run with 1,000 replicates and a MaxTrees value of 10,000. In all cases, the $p$-values between the two sets of tests differed by < 0.015, and therefore the tests with original $p$-values > 0.2 would be extremely unlikely to change the result if the more extensive tests had been run on them. All other settings used for the tests were the PAUP* defaults. Conclusions were drawn at $\alpha = 0.05$. The null hypothesis was that there was no incongruence between the three partitions.

## Approximately unbiased tests

Maximum likelihood trees were generated for all gene regions by individual site (all possible combinations of ($cob$, psbA$^{ncr}$, ITS2) and (BBR, BHB, BLS, BSP, HEW, LIE, LIW); that is, 21 different trees). Trees were also made for each gene region for Atauro Island and Timor datasets (i.e. six trees). The appropriate evolutionary model was determined for each of the 27 datasets individually by first making a neighbour-joining tree using a Jukes-Cantor distance measure and running the `automodel` command. The appropriate evolutionary model for each dataset was then employed when making the ML trees (Table S1). A basic heuristic search was run to generate a base tree or trees, which was then bootstrapped. All bootstrapping procedures used a heuristic search with random sequence addition and had unlimited MaxTrees; all other settings were the PAUP* defaults. $cob$ datasets had 1,000 bootstrap replicates, while the ITS2 and psbA$^{ncr}$ datasets had 100 replicates. In addition, for the psbA$^{ncr}$ datasets, the number of addition sequence replicates was set to 2 ($vs.$ the default of 10), to limit computational burden. The exception is the Atauro Island and Timor datasets, which had 1,000 replicates using the `faststep` search option for all three gene regions. Nodes with <50% bootstrap support were collapsed into polytomies. This procedure yielded 27 ML trees, one for each gene region for each of the nine datasets.

A set of 100 random trees was also generated for each dataset, using the `generate random` command employing an equiprobable model. These additional trees are necessary to gain an accurate $p$-value. In theory, every single possible tree topology of the data should be present, to ensure that the 'true' ML tree is available to be chosen by the test, and to allow calculation of the null distribution for the test statistic (*Goldman, Anderson & Rodrigo, 2000*; *Planet, 2006*). However, given that the number of possible topologies increases exponentially with the addition of taxa, this criterion is functionally impossible to meet for most modern studies. As such, a random subset of all possible topologies is chosen instead (*Robinson et al., 2005*).

Because the AU test assesses whether competing trees are equally likely hypotheses of the data, the choice of dataset will affect the conclusions of the test: it may be expected that for dataset X, tree $T_X$ made from that dataset may be statistically better than another tree $T_Y$, even if they do not inherently disagree. This would not be evidence for incongruence, just the test behaving in its originally intended manner. Because of this, for

each site, reciprocal AU tests were run. For example, for site BBR, the *cob* BBR alignment was initially used as the dataset for the test, and all three trees (from the *cob* BBR, ITS2 BBR and psbA$^{ncr}$ BBR alignments) were compared with the AU test, to see: (a) which of the trees explained the dataset best, and (b) whether the other trees were significantly worse than the best tree at explaining the dataset. This was then repeated using the ITS2 BBR and psbA$^{ncr}$ BBR alignments as the dataset in question, to compare the same three trees. 10,000 resampling estimated log-likelihood (*Kishino, Miyata & Hasegawa, 1990*) bootstrap replicates were used for calculation of *p*-values. Because there were six pairwise comparisons carried out for each site (best tree *vs.* other two trees for *cob*, psbA$^{ncr}$ and ITS2 regions), a within-site Bonferroni correction was applied ($\alpha = 0.0085$). The null hypothesis was that the two trees being compared explained the sequence alignment equally well. The gene regions were considered incongruent if there was reciprocal incongruence; for example, if the ITS2 tree was significantly worse than the *cob* tree at explaining the *cob* dataset and the *cob* tree was significantly worse than the ITS2 tree at explaining the ITS2 dataset.

## Post hoc analyses

Based on the original analyses, several datasets displayed consistent evidence of incongruence (see Results). To verify these results, further ILD tests were executed, using only two gene regions at a time (e.g. for a single site, the following concatenations were assembled and tested: *cob vs.* ITS2; *cob vs.* psbA$^{ncr}$; ITS2 *vs.* psbA$^{ncr}$). As there were three tests *per* site, conclusions were drawn at a Bonferroni-corrected $\alpha = 0.017$. This allowed the location of incongruence to be established (in terms of between gene regions), as the original ILD tests could not say which partitions were incongruent, only that incongruence existed. An extra site which had consistently shown no evidence of incongruence (LIW) was used as a control.

Following that, the datasets which continued to show incongruence had their ML trees input into Dendroscope 3.0 (*Huson & Scornavacca, 2012*), and pairwise tanglegrams were constructed to identify the source of incongruence. In addition, tree hybridisation networks were created using the Autumn algorithm (*Huson & Linz, 2016*), implemented in Dendroscope 3.0. This algorithm attempts to make a consensus tree from two input trees, and identifies the taxa that cannot be reconciled. Finally, raw sequence alignments were inspected to verify incongruence in the identified samples. All background sequences from populations related to putative hybrids were also inspected in an effort to assess the likelihood of possible alternate explanations.

Patterns of putative hybridisation (see Results) could potentially be verified by inspecting an additional non-intragenomically variable nuclear marker. Therefore, the *actin* gene (symbiont nuclear DNA) was sequenced for putative hybrid samples and closely related samples identified in this study. Samples were PCR-amplified and directly sequenced in the forward direction by the Macrogen Sequencing Service (Macrogen Inc., Seoul, South Korea) using the primer pair actin_f1/actin_r1 (*Pochon et al., 2012*). An initial PCR run used a 7 min denaturation at 95 °C, followed by 40 cycles of 94 °C (40 s), 58 °C (40 s), 72 °C (90 s) and a final denaturation of 10 min at 72 °C.
PCRs contained $1 \times$ MyTaq HS Red Mix (Bioline, Randolph, MA, USA), ~ 20 ng sample DNA, 10 µg BSA, 0.25 µM each primer, and $H_2O$ to a total volume of 20 µl. All samples had multiple bands present (observed by running on a 1.5% agarose gel), so the PCR product was run on a 1% agarose gel for 1 h 30 min. Bands at the correct length (~ 900 bp) were excised with a pipette tip and reamplified using 20 cycles of the above conditions. Prior to sequencing, the samples were purified with MagNA PCR clean-up solution (*Rohland & Reich, 2012*). Sequences were inspected visually, and no further steps were taken (see Discussion).

# RESULTS

## Incongruence length difference tests

The *cob*, psbA[ncr] and ITS2 gene partitions for Timor sites (HEW, LIE, LIW) were not incongruent, a trend which was also seen in the overall Timor Island analysis (ILD test, $p = 1$ for all). This $p$-value is not concerning; it simply indicates that among the replicates, the partitions were never more parsimonious than random partitions. The Atauro dataset as a whole did not show statistically substantiated evidence of incongruence, though it approached significance ($p = 0.0874$). In this case, it is valid to use the term 'approaching significance', as the test statistic is directly correlated to the number of replicates for which the original partitions were found to vary from random data (*Planet, 2006*). Looking at each Atauro site individually, BBR and BLS were not incongruent ($p = 0.99$), while BHB displayed an equivocal result ($p = 0.129$) and BSP was strongly incongruent between partitions ($p = 0.001$). However, these tests on three partitions could not identify where potential incongruences were located.

## Approximately unbiased tests

In 24 of the 27 tests conducted, the best tree chosen was the one that was made from that gene region originally (i.e. for a test with the ITS2 region as its base, the ITS2 tree was chosen as the best tree). The exceptions were sites BBR, HEW and LIE, where either the psbA[ncr] or ITS2 trees were chosen as the best explanation of the *cob* dataset. The test always found incongruence when using the psbA[ncr] region as a base; this is likely due to an issue with the test (see Discussion), and therefore the results for the *cob* and ITS2 gene regions are the major focus of these results.

There was a very clear island-wide partitioning of results when it came to the AU test (Table 1). All Timor sites (HEW, LIE, LIW) were not incongruent for either the *cob* or ITS2 gene regions; all three trees (*cob*, ITS2, psbA[ncr]) did an equally good job of explaining these two regions. While there was incongruence between *cob* and ITS2 trees using the ITS2 region as a base in the overall Timor analysis, this was not reciprocated (i.e. these two trees were not incongruent when considering the *cob* dataset). In contrast, the Atauro datasets showed high levels of reciprocal incongruence. Overall, the ITS2 tree (but not the psbA[ncr] tree) made from all Atauro samples was incongruent with the *cob* dataset, and both the *cob* and psbA[ncr] trees were incongruent with the ITS2 dataset. Looking at individual sites, the same complete reciprocal incongruence exists for the BHB and BSP datasets. These three datasets (Atauro, BHB, BSP) correspond to the three lowest $p$-values
**Table 1  Results of the approximately unbiased (AU) tests.**

| Dataset | Gene region used for test | Best tree | Tree to compare with best tree | AU $p$-value |
|---|---|---|---|---|
| BBR | $cob$ | psbA$^{ncr}$ | ITS2 | <0.0001** |
| | | | $cob$ | 0.4417 |
| | ITS2 | ITS2 | $cob$ | 0.4056 |
| | | | psbA$^{ncr}$ | 0.7712 |
| | psbA$^{ncr}$ | psbA$^{ncr}$ | $cob$ | <0.0001** |
| | | | ITS2 | <0.0001** |
| BHB | $cob$ | $cob$ | ITS2 | <0.0001* |
| | | | psbA$^{ncr}$ | 0.5631 |
| | ITS2 | ITS2 | $cob$ | <0.0001* |
| | | | psbA$^{ncr}$ | <0.0001* |
| | psbA$^{ncr}$ | psbA$^{ncr}$ | $cob$ | <0.0001** |
| | | | ITS2 | <0.0001** |
| BLS | $cob$ | $cob$ | ITS2 | <0.0001* |
| | | | psbA$^{ncr}$ | 0.3456 |
| | ITS2 | ITS2 | $cob$ | 0.4163 |
| | | | psbA$^{ncr}$ | 0.1806 |
| | psbA$^{ncr}$ | psbA$^{ncr}$ | $cob$ | <0.0001** |
| | | | ITS2 | <0.0001** |
| BSP | $cob$ | $cob$ | ITS2 | 0.0493 |
| | | | psbA$^{ncr}$ | <0.0001* |
| | ITS2 | ITS2 | $cob$ | <0.0001* |
| | | | psbA$^{ncr}$ | <0.0001* |
| | psbA$^{ncr}$ | psbA$^{ncr}$ | $cob$ | <0.0001** |
| | | | ITS2 | <0.0001** |
| HEW | $cob$ | psbA$^{ncr}$ | ITS2 | 0.1562 |
| | | | psbA$^{ncr}$ | 0.1562 |
| | ITS2 | ITS2 | $cob$ | 0.5465 |
| | | | psbA$^{ncr}$ | 0.5465 |
| | psbA$^{ncr}$ | psbA$^{ncr}$ | $cob$ | <0.0001** |
| | | | ITS2 | <0.0001** |
| LIE | $cob$ | ITS2 | $cob$ | 0.0183 |
| | | | psbA$^{ncr}$ | 0.2336 |
| | ITS2 | ITS2 | $cob$ | 0.0870 |
| | | | psbA$^{ncr}$ | 0.4727 |
| | psbA$^{ncr}$ | psbA$^{ncr}$ | $cob$ | <0.0001** |
| | | | ITS2 | <0.0001** |
| LIW | $cob$ | $cob$ | ITS2 | 0.0409 |
| | | | psbA$^{ncr}$ | 0.0811 |
| | ITS2 | ITS2 | $cob$ | 0.2490 |
| | | | psbA$^{ncr}$ | 0.6638 |
| | psbA$^{ncr}$ | psbA$^{ncr}$ | $cob$ | <0.0001** |
| | | | ITS2 | <0.0001** |

| Table 1 (continued). | | | | |
| --- | --- | --- | --- | --- |
| Dataset | Gene region used for test | Best tree | Tree to compare with best tree | AU $p$-value |
| Atauro | cob | cob | ITS2 | <0.0001* |
| | | | psbA$^{ncr}$ | 0.0125 |
| | ITS2 | ITS2 | cob | <0.0001* |
| | | | psbA$^{ncr}$ | <0.0001* |
| | psbA$^{ncr}$ | psbA$^{ncr}$ | cob | <0.0001** |
| | | | ITS2 | <0.0001** |
| Timor | cob | cob | ITS2 | 0.5604 |
| | | | psbA$^{ncr}$ | 0.0196 |
| | ITS2 | ITS2 | cob | <0.0001* |
| | | | psbA$^{ncr}$ | 0.0935 |
| | psbA$^{ncr}$ | psbA | cob | <0.0001** |
| | | | ITS2 | <0.0001** |

**Note:**
$P$-values presented are whether a candidate tree is statistically differentiable from the best tree. Statistical significance is designated by * (conclusions drawn at Bonferroni-corrected $\alpha$ = 0.0085); $p$-values likely due to type I error are designated by ** (see Discussion).

returned by the ILD tests. In general, the tests reveal incongruence between the organellar (cob and psbA$^{ncr}$) and nuclear (ITS2) gene regions. In all cases, the AU test was unable to reject congruence between the cob and psbA$^{ncr}$ regions. However, it did reject congruence between the ITS2 and psbA$^{ncr}$ regions (using the ITS2 region as a base), and showed reciprocal incongruence between the cob and ITS2 region (using both the ITS2 and cob regions as a base). In addition, the ITS2 tree was incongruent with the cob dataset (but not the other way around) for site BLS. As such, these four datasets (complete Atauro, BHB, BSP, BLS) were carried forward to post hoc testing.

## Post hoc analyses

Additional ILD tests were carried out using two partitions at a time. Site LIW was included as a control to ensure that the tests still successfully supported congruence where appropriate. These results strongly support the AU test (Table 2). There is clear incongruence between the nuclear ITS2 region and the other two organellar gene regions, which are not incongruent when considered together. Site LIW is clearly not incongruent at all regions. This shows that these two-way tests are functioning as expected. BLS is also not incongruent; while the AU test indicated potential incongruence, the other tests do not and so it was not carried forward as a candidate for hybridisation. Pairwise tanglegrams were made for BHB, BSP and Atauro datasets, with potentially incongruent branches verified by attempting to hybridise the two trees to create a consensus. Those branches and closely related sequences subsequently had their raw sequences inspected in an attempt to confirm incongruence.

The tanglegrams and tree hybridisation analyses for site BHB (Fig. 2) support the results of the statistical tests. Comparing the two organellar genes with the ITS2 region (Figs. 2A and 2B) reveals two incongruent samples, BHB146 and BHB148, while BHB148 is also

**Table 2  Results of pairwise incongruence length difference tests.**

| Dataset | Partitions tested | $p$-value |
|---|---|---|
| BHB | *cob vs.* ITS2 | 0.006* |
|  | *cob vs.* psbA$^{ncr}$ | 0.847 |
|  | ITS2 *vs.* psbA$^{ncr}$ | 0.021 |
| BSP | *cob vs.* ITS2 | 0.011* |
|  | *cob vs.* psbA$^{ncr}$ | 0.223 |
|  | ITS2 *vs.* psbA$^{ncr}$ | 0.001* |
| Atauro | *cob vs.* ITS2 | 0.01* |
|  | *cob vs.* psbA$^{ncr}$ | 1 |
|  | ITS2 *vs.* psbA$^{ncr}$ | 0.01* |
| BLS | *cob vs.* ITS2 | 1 |
|  | *cob vs.* psbA$^{ncr}$ | 1 |
|  | ITS2 *vs.* psbA$^{ncr}$ | 0.778 |
| LIW | *cob vs.* ITS2 | 1 |
|  | *cob vs.* psbA$^{ncr}$ | 1 |
|  | ITS2 *vs.* psbA$^{ncr}$ | 1 |

**Note:**
Conclusions were drawn at a Bonferroni-corrected $\alpha$ = 0.017. Statistical significance is designated by *.

incongruent between the *cob* and psbA$^{ncr}$ regions (Fig. 2C). Inspection of raw sequence alignments reveals BHB146 is an example of true incongruence (Fig. 3), whereas the incongruence in BHB148 is due to a highly divergent psbA$^{ncr}$ sequence, and does not show a reticulate pattern (Fig. S1). For the organellar gene regions, BHB146 belongs to the *Cladocopium* C1 radiation (symbiont types C42a and C1v respectively, see *Brian, Davy & Wilkinson, 2019*). For the ITS2 region, it is identified as type C1#, which groups more closely with the *Cladocopium* C3 radiation.

The BSP tanglegrams (Fig. 4) also support the statistical analyses, with six potentially incongruent samples identified. After inspection of the raw sequence data, four of these were disregarded (BSP211, BSP320, BSP372, BSP387), as they were more likely caused by parsimony-uninformative mutations in a single sequence (Fig. S2). However, two samples were verified as incongruent (BSP343 and BSP364, Figs. 5 and 6). BSP364 belongs to two different previously defined subclades: a variant of *Cladocopium* type C40 for psbA$^{ncr}$, and type C3z for ITS2. BSP343 also shows clear incongruence between the organellar and nuclear genes regions. The psbA$^{ncr}$ is a variant of *Cladocopium* type C40, which groups it most closely with the psbA$^{ncr}$ C3z subclade (Fig. 4B), while the ITS2 region features type C3u, which places it as distinct from both the C3z and C40 groups.

Pairwise tanglegrams and hybridisation analyses were also executed for the whole Atauro Island dataset (sites BBR, BHB, BLS, BSP). Despite the inclusion of two more sites, the analyses showed that incongruence was caused by exactly the same samples as found by the individual site analyses, affirming BHB and BSP as sites with incongruent samples. Further, no other sites contributed any incongruent samples. The overall results are presented in Table 3, which demonstrates that ITS2 comparisons displayed the most incongruence, while any incongruences between *cob* and psbA$^{ncr}$ regions were due to

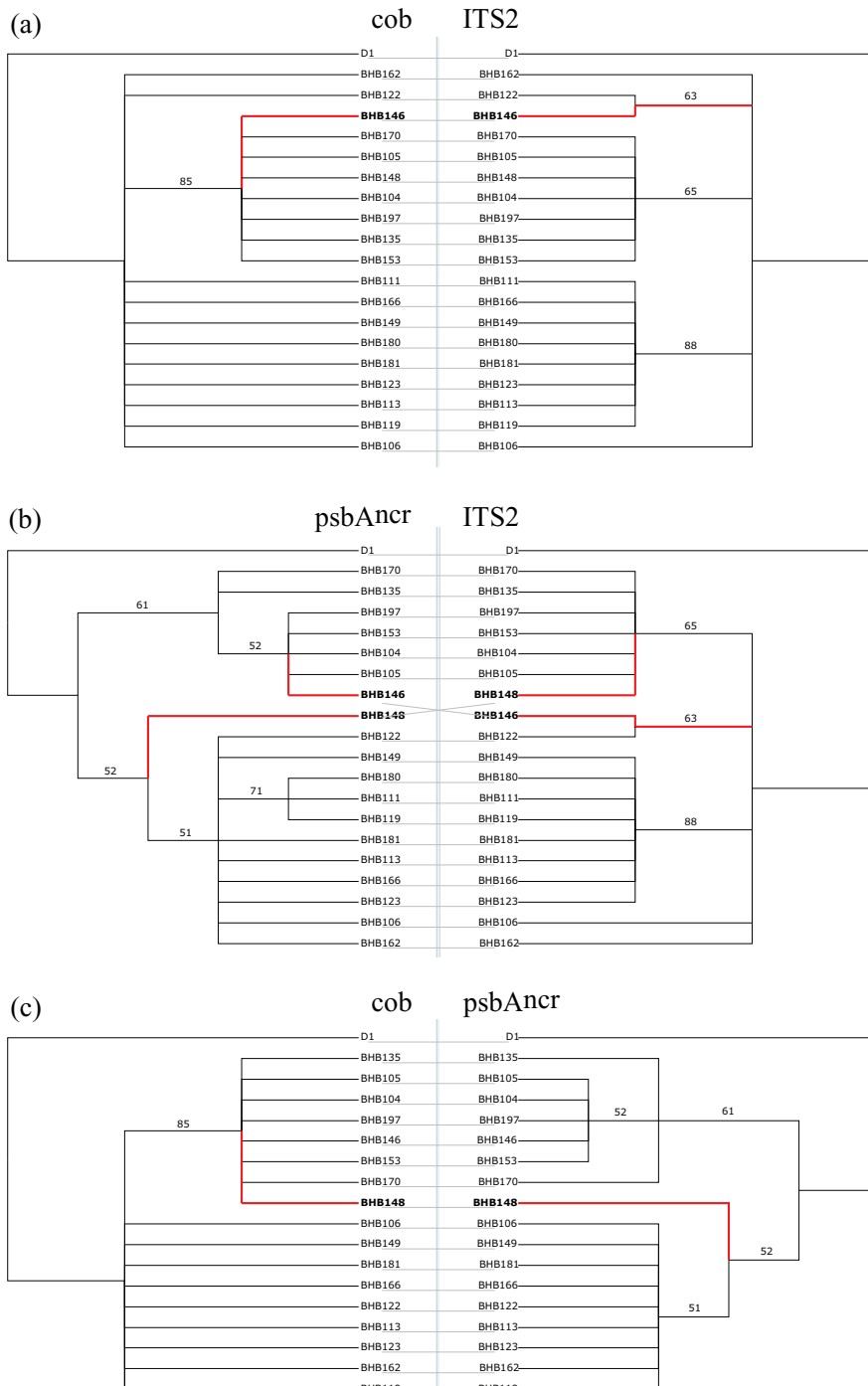

**Figure 2 Pairwise tanglegrams for site BHB.** Red branches with bolded taxa labels indicate incongruent samples, as identified by the tree hybridisation analysis (executed in Dendroscope 3.0 (*Huson & Linz, 2016*)). Branch labels are ML bootstrap values (1,000 replicates for *cob*, 100 for ITS2 and psbA^ncr). (A) *cob vs.* ITS2: found incongruent by ILD and AU tests. (B) psbA^ncr *vs.* ITS2: found incongruent by ILD and AU tests. (C) *cob vs.* psbA^ncr: found congruent by ILD and AU tests.

```
                              310     315           455
                               |       |             |
(a) cob     BHB104   GGGAGTAC  ...  TTCTT
            BHB105   GGGAGTAC  ...  TTCTT
            BHB122   GGGGGTAC  ...  TTGTT
            BHB146   GGGAGTAC  ...  TTCTT
            BHB149   GGGGGTAC  ...  TTGTT

                                    225     230    235
                                     |       |      |
(b) ITS2    BHB104   AGG--TTTCTACCTTCGTG
            BHB105   AGG--TTTCTACCTTCGTG
            BHB122   AAG--TTTCTACCTTCGCG
            BHB146   AAG--TTTCTACCTTCGCG
            BHB149   AAG--TTTCTACCTTCGTG

                               270    275    280    285
                                |      |      |      |
(c) psbAncr  BHB104  CCCTTCGGG-GTGCACAT
             BHB105  CCCTTCGGG-GTGCACAT
             BHB122  CCCGTAGGG-GTACCCAT
             BHB146  CCCTTCGGG-GTGCACAT
             BHB149  CCCGTAGGG-GTACCCAT
```

**Figure 3 Short selections of raw sequence data for incongruent sample BHB146 and related sequences (polymorphisms in bold).** In organellar gene regions (A) and (C), BHB146 groups with samples BHB104 and BHB105 (*Cladocopium* type C42a, C1v). In the nuclear gene region (B), BHB groups with BHB122 and BHB149 (*Cladocopium* type C1#).

non-reticulate sequence variation. This is strongly supportive of the AU test results as well as Table 2, which all indicate that incongruence occurs between the organellar and nuclear genomes of *Cladocopium*. Of the three clearly incongruent samples (BHB146, BSP343, BSP364) there was no general clear pattern in coral host (host genera: *Pavona, Symphyllia* and *Acropora*, respectively). The sequencing of the *actin* gene was uninformative, with only occasional non-parsimonious variation observed (i.e. polymorphisms in a single sequence only).

Background sequences for populations related to the putative hybrids were also analysed. This was particularly fruitful for sample BSP364 (which has incongruent C3z and C40 genetic signals). At site BSP, there were eight additional samples with C3z as the dominant ITS2 sequence (these were also C3z for psbA$^{ncr}$ and *cob* regions), and thirteen with C40 as the dominant ITS2 sequence (these were also C40 for the psbA$^{ncr}$ and *cob* regions) (Fig. 7). Of the eight C3z samples (Fig. 7A), seven had no C40 sequences in their genomes, while one had C40 traces at a frequency of 0.61% (Fig. 7B). C40 samples had a

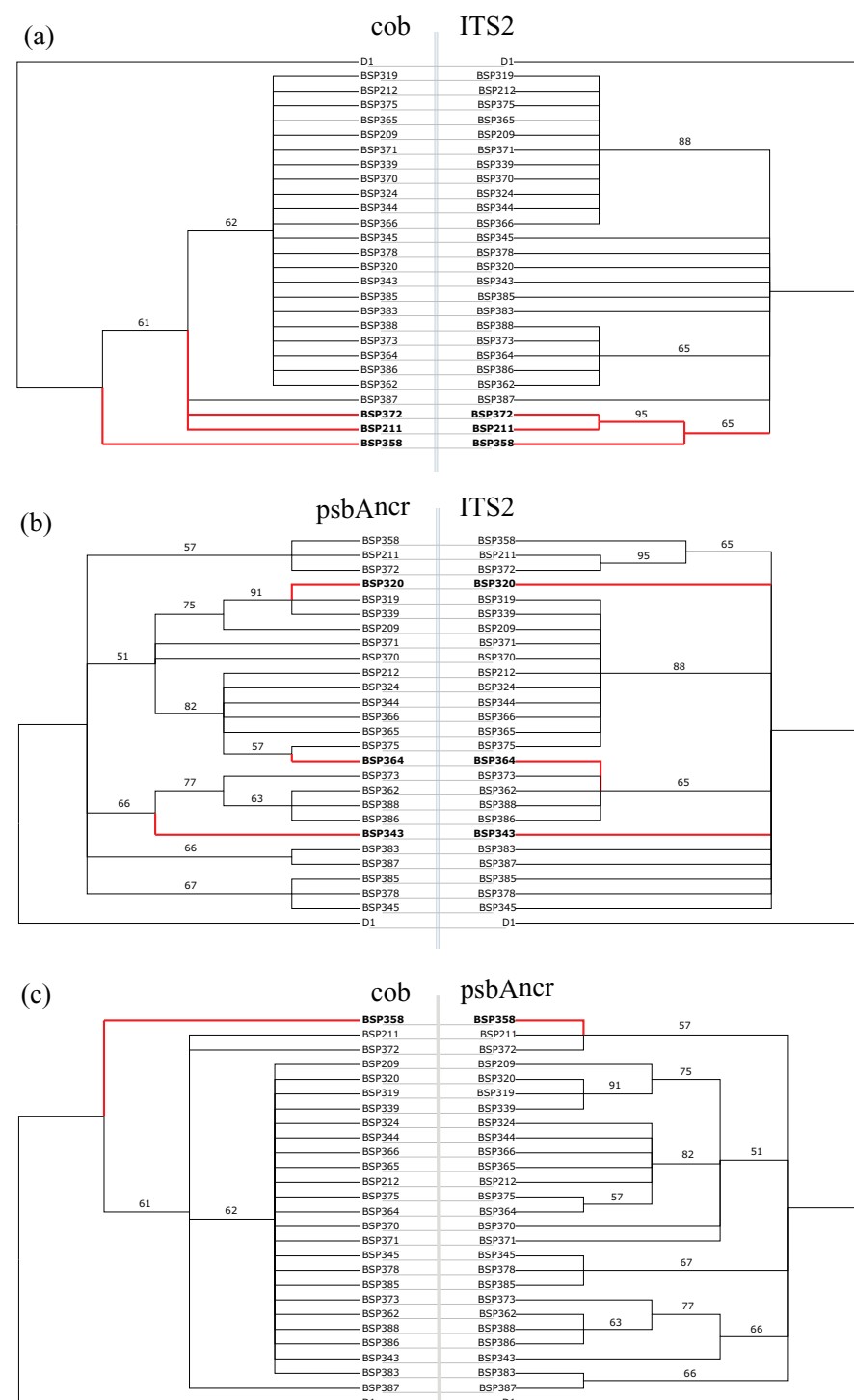

**Figure 4 Pairwise tanglegrams for site BSP.** Red branches with bolded taxa labels indicate incongruent samples, as identified by the tree hybridisation analyses (executed in Dendroscope 3.0 (*Huson & Linz, 2016*)). Branch labels are ML bootstrap values (1,000 replicates for *cob*, 100 for ITS2 and psbA^ncr). (A) *cob vs.* ITS2: found incongruent by ILD and AU tests. (B) psbA^ncr *vs.* ITS2: found incongruent by ILD and AU tests. (C) *cob vs.* psbA^ncr: found congruent by ILD and AU tests.

```
                          205    210         290
                           |      |           |
(a) ITS2    BSP343    TG-CGCGC  ...  CCGCT
            BSP383    TG-CGCGC  ...  CCGCT
            BSP386    TG-TGCGC  ...  CTGCT
            BSP387    TG-CGCGC  ...  CCGCT
            BSP388    TG-TGCGC  ...  CTGCT
```

```
                             425    430   435
                              |      |     |
(b) psbA^ncr  BSP343   ATGCC-CCACA-GGGGCAT
              BSP383   ACACC-CCGGA-GGGGTGT
              BSP386   ATGCC-CCACA-GGGGCAT
              BSP387   ACACC-CCGGA-GGGGTGT
              BSP388   ATGCC-CCACA-GGGGCAT
```

**Figure 5 Short selections of raw sequence data for incongruent sample BSP343 and related sequences (polymorphisms in bold).** (A) In the nuclear ITS2 region, BSP343 groups with samples BSP383 and BSP387; point mutations at base pairs 23 and 238 (available in Data Availability) identify it as *Cladocopium* type C3u. (B) In the organellar psbA^ncr region, BSP343 groups with BSP386 and BSP388, as a variant of *Cladocopium* type C40. The *cob* gene was invariant in this case.

```
                         255    260    265
                          |      |      |
(a) ITS2    BSP344    TGCTTGCGACCGCTGG
            BSP362    TGCTTGCAACTGCTGG
            BSP364    TGCTTGCAACTGCTGG
            BSP366    TGCTTGCGACCGCTGG
            BSP373    TGCTTGCAACTGCTGG
```

```
                         310    315    320
                          |      |      |
(b) psbA^ncr  BSP344   ATG-AAAAGAAAAAGA
              BSP362   ACGAAAAAGAAAAATA
              BSP364   ATG-AAAAGAAAAAGA
              BSP366   ATG-AAAAGAAAAAGA
              BSP373   ACGAAAAAGAAAAATA
```

**Figure 6 Short selections of raw sequence data for incongruent sample BSP364 and related sequences (polymorphisms in bold).** (A) In the nuclear ITS2 region, BSP364 groups with samples BSP362 and BSP373 (*Cladocopium* type C3z). (B) In the organellar psbA^ncr region, BSP364 groups with BSP344 and BSP366, as a variant of *Cladocopium* type C40. The *cob* gene was invariant in this case.

**Table 3 Summary of incongruent samples inferred from tanglegrams and tree hybridisation analyses.**

| Dataset | Comparison | Incongruent Samples |
|---|---|---|
| BHB | *cob vs.* ITS2 | **BHB146** |
| | psbA$^{ncr}$ *vs.* ITS2 | **BHB146**, BHB148 |
| | *cob vs.* psbA$^{ncr}$ | BHB148 |
| BSP | *cob vs.* ITS2 | BSP211, BSP358, BSP372 |
| | psbA$^{ncr}$ *vs.* ITS2 | BSP320, **BSP343**, **BSP364** |
| | *cob vs.* psbA$^{ncr}$ | BSP358 |
| Atauro | *cob vs.* ITS2 | **BHB146**, BSP372, BSP387 |
| | psbA$^{ncr}$ *vs.* ITS2 | **BHB146**, BHB148, **BSP343**, **BSP364**, BSP372 |
| | *cob vs.* psbA$^{ncr}$ | BHB148, BSP372, BSP387 |

**Note:**
Bolded samples are those verified to be incongruent.

low proportion of background sequences (Fig. 7C), and less than half of them possessed C3z traces, which only appeared as the 3rd most common background sequence in terms of average abundance, and 5th most common in terms of frequency (Fig. 7D). The other two putative hybrid ITS2 types (C3u, C1#) did not occur frequently enough to conduct a similar analysis, but are mentioned in the Discussion.

# DISCUSSION

## Methodological approach taken

There are many factors, such as character sampling and bias due to differential gene length, which can give false signals of incongruence (*Som, 2014*). However, the approach taken in this study has been able to clearly display incongruence between organellar and nuclear regions in *Cladocopium*. In isolation, it is true that there are issues with the tests utilised. For example, the AU test presented an issue with most trees being incongruent for the psbA$^{ncr}$ region. The psbA$^{ncr}$ region is highly variable (*LaJeunesse & Thornhill, 2011*; *Thornhill et al., 2014*), and hence a more complex tree is required to explain it. The *cob* and ITS2 trees with multiple polytomies could not do this as effectively, and hence a result of incongruence was returned. Therefore, the results from the *cob* and ITS2 datasets are likely more reliable, and were the focus of the Results. Further, the ILD test has been criticised for being overly sensitive, especially when comparing partitions of different resolutions (*Barker & Lutzoni, 2002*). The refutation of this is simple: in all cases, it found congruence between the psbA$^{ncr}$ and *cob* regions, the two most different in terms of resolution (Table 2), so this is clearly not contributing to the positive results between the organellar and nuclear partitions observed here. Indeed, it failed to reject congruence between the *cob* and psbA$^{ncr}$ regions for sites BSP and BHB despite the tree hybridisation analyses finding potential incongruence (Figs. 2C and 4C), and so appears to be reasonably conservative in this case. The results of the AU and ILD tests are also compelling because they are differential: they show consistently different patterns between datasets and are therefore likely responding to genuine phylogenetic signals. This was confirmed by looking at the raw sequence data, and shows the efficacy of the approach taken here.

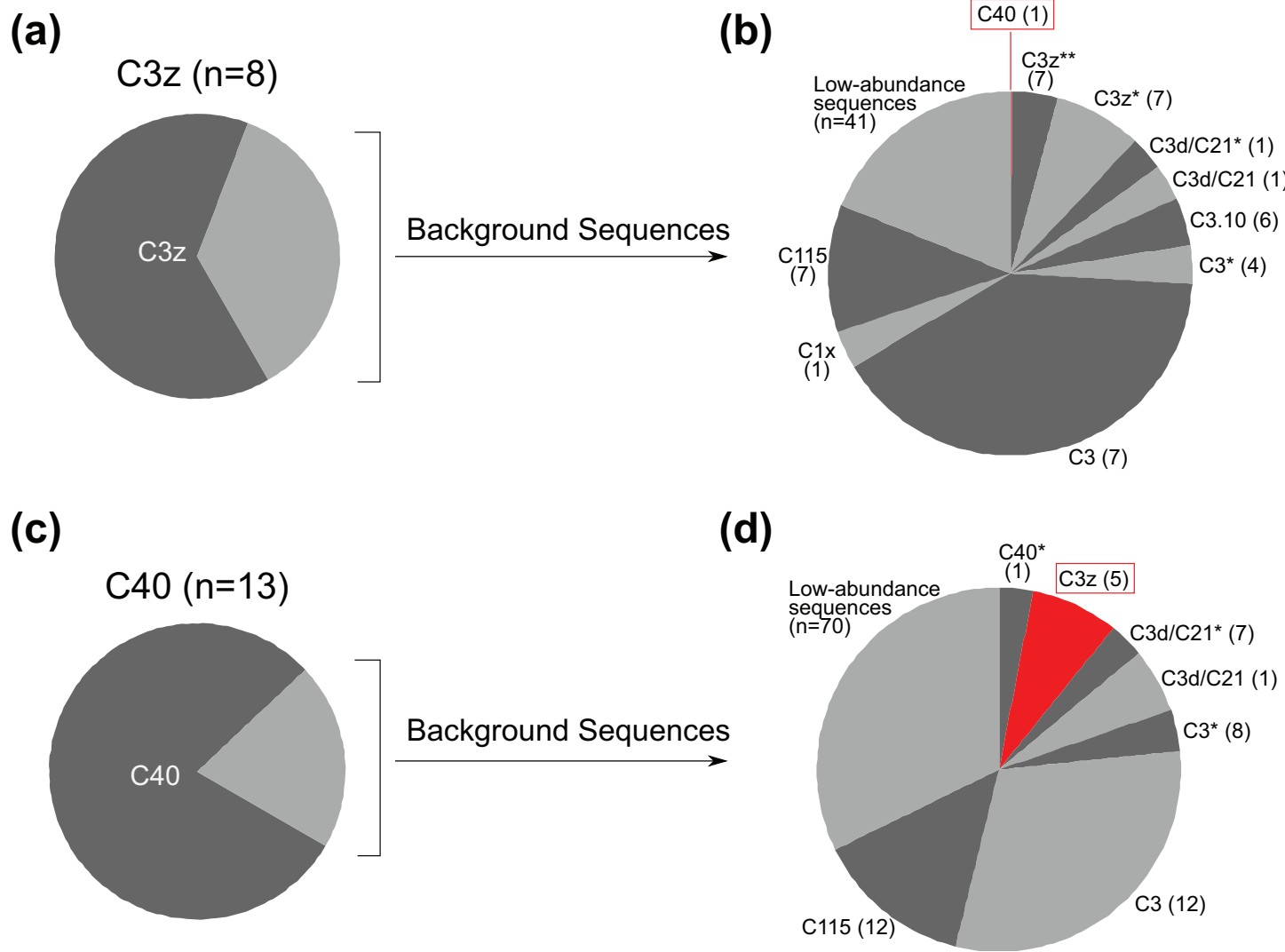

**Figure 7 Average proportions of different background sequence populations for samples identified as C3z (eight samples) and C40 (13 samples) from site BSP.** (A) For C3z population, proportion of sequences that were C3z *vs.* non-C3z sequences (i.e. background sequences), averaged over all eight samples. (B) Average proportion of background sequences for C3z samples, identified using the database of *Franklin et al. (2012)* and GenBank BLAST search. Sequences followed by stars (* or **) indicate novel sequences and are named according to their most closely related sequence in the databases. Parentheses indicate the total number of samples (out of eight) that the background sequence appeared in. Rare sequences (<3% average individual abundance) are clustered together. The red box indicates the background sequence with the organellar identity of the putative hybrid (BSP364). (C) For C40 population, proportion of sequences that were C40 *vs.* non-C40 sequences (i.e. background sequences), averaged over all 13 samples. (D) Average proportion of background sequences for C40 samples. Parentheses indicate the total number of samples (out of 13) that the background sequence appeared in. The red box indicates the background sequence with the nuclear identity of the putative hybrid. Other details as per (B).

With such a wide range of samples, initially searching for incongruences in sequence data would be functionally impossible, as it would require comparing all possible combinations of sequences (in this study, this would require $1.17 \times 10^{278}$ comparisons). However, the stepwise use of analyses allowed the initial identification of which sites may host incongruent samples, and then visualisation on phylogenetic trees allowed simple alignments of appropriate samples to be generated, where incongruence could clearly be

refuted or confirmed. In addition, given the issues with tests in isolation, the multiplicity of analyses used generates a far more convincing picture of reticulate evolution.

## Hybridisation in *Cladocopium*?

Incongruence was comprehensively established for the samples BHB146, BSP343 and BSP364. However, this does not necessarily translate to hybridisation, as there are a range of analytical or biological factors that can cause incongruence in phylogenetic data. For example, one hypothesised to be quite common but insidious in its undetectable nature is heterotachy, shifts in site-specific evolutionary rates through time (*Som, 2014*). While there is no particular way to identify heterotachy or exclude it as a cause, except with a very large number of sequences, ML methods in particular have been shown to be robust to even intermediate levels of heterotachy (*Som, 2014*).

A more plausible explanation is ILS, often considered the most common cause of incongruence (*Degnan & Rosenberg, 2009*). This is due to polymorphisms not segregating fully during speciation events, leading to phylogenetic signals in gene trees that conflict with the overall species tree. This has been shown to be quite common in the ITS2 region, thanks to its multiple-copy nature (*Thornhill, LaJeunesse & Santos, 2007*). Through this mechanism, ancestral polymorphisms may persist at low levels in the genome. Therefore, it is possible that the divergent sequences recovered actually represent a single symbiont population, which has multiple ancestral polymorphisms present via ILS (i.e. intragenomic variation). Through stochastic DNA processes such as unequal crossing over, slipped-strand mispairing and transposition, these intragenomic variants may be eliminated or promoted in the multiple-copy array (i.e. concerted evolution, see *Nei & Rooney, 2005*). Hence, in the samples from a single reproductively isolated population, one ancestral polymorphism may be dominant in the ITS2 region of some, while a different ancestral polymorphism may be dominant in others. This would cause the patterns observed in this study, with the ITS2 region being occasionally incongruent with the organellar regions.

Ideally, a statistical test would be carried out to differentiate between hybridisation and ILS, and such tests do exist. However, they require inputs of information which are not currently available for *Cladocopium*, such as: (a) An understanding of the effective population size $N_e$ (*Pelser et al., 2010*); (b) a large number of genes, at least some of which must be adjacent (*Pollard et al., 2006*; *Meng & Kubatko, 2009*); or (c) strictly bifurcating trees and clearly defined species (*Sang & Zhong, 2000*; *Joly, McLenachan & Lockhart, 2009*). Therefore, ILS as a cause of the observed incongruence cannot be statistically refuted. However, there is good evidence that the patterns observed here are more likely to be caused by symbiont hybridisation.

First, the pattern of incongruence observed, with organellar cytoplasmic genes being different to nuclear genes, accords with a large body of prior theory on hybridisation. Nuclear genes are largely inherited biparentally, and the ITS2 region is no exception (*Baldwin et al., 1995*; *Rybalka et al., 2013*). However, the cytoplasm tends to be inherited maternally (*Rieseberg, Whitton & Linder, 1996*). This difference is largely due to gametogenesis and fertilisation, where the male gamete typically only contains nuclear

information, while the female gamete (egg) contains the cytoplasm that will be passed on to the zygote. Therefore, if an organism encounters a population of another species and produces viable hybrids, theory predicts that over time, repeated backcrossing with the more common species (introgression) will produce hybrids with divergent organellar and nuclear signals. While the nature of the sexual life cycle has yet to be fully elucidated in the Symbiodiniaceae, previous evidence has shown that other unicellular dinoflagellates produce gametes (*Brawley & Johnson, 1992*). In addition, the presence of 'plus' and 'minus' mating types, analogous to gender, has been shown in the dinoflagellate *Alexandrium tamarense* (*Brosnahan, 2011*). Therefore, it is reasonable to assume that Symbiodiniaceae also produce distinct gametes (as opposed to conducting sex via fusion, for example), making this mechanism eminently plausible. The documentation of functional meiotic genes in Symbiodiniaceae (*Chi, Parrow & Dunthorn, 2014*; *Levin et al., 2016*) supports this assertion. Such a pattern of discordance between cytoplasmic and nuclear genes caused by hybridisation has been recorded for taxa as diverse as plants (*Rieseberg, Whitton & Linder, 1996*; *Pelser et al., 2010*; *Sun et al., 2015*), beetles (*Sota & Vogler, 2001*) and indeed corals (*Van Oppen et al., 2001*). In general, hybridisation is predicted to cause incongruence between nuclear and cytoplasmic markers in both multicellular and unicellular taxa (*Bull et al., 1993*). Other factors due to hybridisation, such as semigamy or differential fitness of nuclear-cytoplasmic combinations, can also cause incongruence between nuclear and cytoplasmic gene trees (*Rieseberg, Whitton & Linder, 1996*). Therefore, the fact that this was the pattern observed in this study is strong circumstantial evidence that hybridisation is the explanation.

In addition, hybridisation is made more likely in comparison to ILS by the fact that all of the incongruent ITS2 sequences were previously defined types (i.e. not unique sequences), that were also present in non-incongruent relationships in the analyses. For example, BSP364 had a generic *Cladocopium* type C3 sequence for the *cob* gene, was a C40 type for the psbA$^{ncr}$ region, and C3z for the ITS2 region. Significantly, there were also samples recovered which were type C40 for both the psbA$^{ncr}$ and ITS2 regions (samples BSP319–BSP375, Fig. 4B), and samples which were type C3z for both regions (samples BSP373–BSP386, Fig. 4B). This confirms that they are clearly separate types, supported by the fact that they differ by four base pairs in the ITS2 sequence and 64 base pairs in the psbA$^{ncr}$ region (including a 49 base pair deletion in the C40 sequences), indicating that this is not just a non-diagnostic polymorphism (*Wilkinson et al., 2015*). The implications for this being caused by ILS are given in Fig. 8. Only the psbA$^{ncr}$ and ITS2 genes are presented, as the *cob* gene was invariant in this case.

Figure 8B graphically represents the process that would be required for the observed patterns to be due to ILS. Given that symbiont sex is now strongly supported (though in low frequency; *Thornhill et al., 2017*), it seems unlikely that a divergent ancestral polymorphism could be maintained as the *dominant* sequence in some samples within type C40, as it would be expected that repeated recombination would eventually remove C3z traces from the C40 genome, or vice versa (Fig. 8A). It is more parsimonious that a hybridisation event has occurred between symbiont types C40 and C3z, with backcrossing leading to incongruence between organellar and nuclear genes. This is

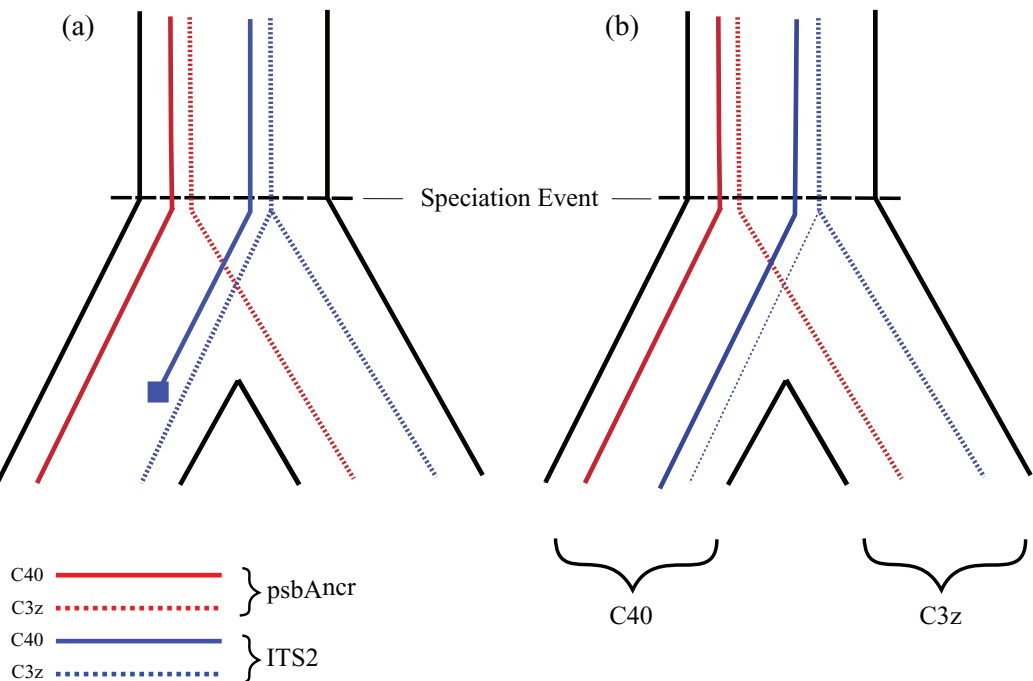

**Figure 8 Predictions under incomplete lineage sorting.** (A) General pattern expected for ILS. A single ancestral population with polymorphism in both the psbA^ncr and ITS2 regions is present before a speciation event. After speciation, the ITS2 polymorphism fails to segregate, while through stochastic processes the C40 polymorphism is eliminated and leads to incongruence between nuclear and chloroplast genes. (B) The process of ILS that would be required for this example. The ITS2 region fails to segregate after speciation; despite the extensive presence of C40 alleles, a small subpopulation of symbionts with dominant C3z alleles is maintained (weak dashed blue line) in the C40 population and both are recovered in present-day sampling, at the same site, as pure C3z populations.

strongly supported by the analysis of the background symbiont populations (Fig. 7). The results show that there is little evidence of C40 and C3z sequences being shared within samples. The C3z population had almost no C40 sequences present at all, with just one sample having an extremely low background abundance of C40 (Fig. 7B). C3z sequences were slightly more common in C40 samples (Fig. 7D). However, the low proportion of background sequences in this population (Fig. 7C) meant that overall the presence of C3z in the C40 population was negligible (mean = 1.61%, median = 0). This reveals essentially pure populations of C40 and C3z at site BSP, something which strongly favours hybridisation *vs.* ILS as causing the mixed pattern in BSP364 (*Wilkinson et al., 2015*). While the other two putative hybrid ITS2 types (C3u, C1#) do not have large populations to compare, the same basic pattern was also observed for BSP343, which was identified as *Cladocopium* type C40 for the organelle regions, and type C3u for ITS2. If this was to be caused by ILS, then *both* variants would be expected to occur in the ITS2 region, (with one at low frequency), but the NGS data revealed no trace of ITS2 type C40 in that sample. Further, the divergences observed (i.e. C40/C3u, C40/C3z, C1/C1#) all coalesce at the 'ancestral' types C1 or C3, rather than one representing an intermediate evolutionary step to the other. Therefore, ILS would also predict these ancestral sequences to be in the

ITS2 genome in low frequencies. However, this was only observed in BSP343 (as the fourth most common sequence); neither BHB146 nor BSP364 showed any evidence of these ancestral sequences. While it is acknowledged that hybridisation and ILS are not mutually exclusive and the incongruences observed could be caused by a combination of both, the weight of evidence suggests that these results are more likely a result of interspecific hybridisation between distinct symbiont types.

Potentially, the two competing hypotheses could be distinguished by sequencing another nuclear gene, less susceptible to intragenomic variation, for both putative hybrid samples and closely related sequences. If the patterns were due to hybridisation, it would be expected that the additional nuclear gene would support the ITS2 identity, and cluster the sample with the same group as presented in the ITS2 trees (Figs. 2 and 4). In contrast, if the incongruence was caused by ILS, the additional marker would cluster the putative hybrid with the same samples as the organellar gene regions. This was attempted using the *actin* gene. Unfortunately, low resolution (and difficulties in amplification leading to short usable sequences) meant that neither scenario was supported, as the sequences were not variable enough to recover the groups observed in Figs. 2 and 4. The other currently-available Symbiodiniaceae nuclear gene markers either suffer from the same issue of significant intragenomic variation (ITS1), or are lower-resolution than *actin* (SSU, LSU, 5.8S, *elf2*), and therefore the patterns observed cannot currently be independently verified. The further development of highly-variable, reliably amplifiable nuclear gene markers should be a priority for Symbiodiniaceae systematics. However, ILS (and indeed all analytical factors), are random or would be expected to affect all sites. The results obtained, however, are anything but random, with two sites consistently being recovered as incongruent in contrast to all others, despite those incongruences coming from a range of host species that were present at all sites. In addition, both these sites have been shown to be rich in Symbiodiniaceae diversity, when compared with the Timor sites (*Brian, Davy & Wilkinson, 2019*). This suggests that putative hybridisation may be limited to high-quality sites that maintain high levels of symbiont diversity.

Intragenomic variation within the ITS2 region could lead to the incongruences observed via ILS, though the discussion above suggests that hybridisation should be favoured as an explanation. However, the psbA$^{ncr}$ region can also be intragenomically variable (*LaJeunesse & Thornhill, 2011*), with intragenomic sequence ratios that may fluctuate within a single species. Combined with the intragenomic variation of the ITS2 region, this generates the potential for a wide range of ITS2-psbA$^{ncr}$ combinations within a single genome. For example, one member of a population may have a 9:1 ratio of variant A: variant B within its multiple-copy ITS2 sequences, and a 9:1 ratio of variant A:variant B in its multiple-copy psbA$^{ncr}$ sequences. A second member of the same population could plausibly have an 8:2 ratio of variant A:variant B within its ITS2 sequences, and a 4:6 ratio of variant A:variant B in its psbA$^{ncr}$ sequences. Therefore, there is a possible difference between the most common *sequences* in total, and the most common *associations* between ITS2 and psbA$^{ncr}$ ratio types. If only the most common sequences are studied, there is the potential for some natural associations (i.e. not caused by hybridisation) to appear as incongruences. The present study attempts to draw conclusions based on common

associations between nuclear and organellar genes, but is only able to utilise common sequences. This is explicit for the psbA$^{ncr}$ region (as Sanger sequencing only amplifies the most common overall sequence), and implicit for the ITS2 region (as the nature of the tests necessitated the selection of the most common overall sequence from NGS data). Within a single genome, a solution would be to sequence multiple markers from the same DNA strand through long-read sequencing, which would preserve the ratio of intragenomic variants. However, this does not work for markers across multiple organellar and nuclear genomes, and currently there is no other acceptable solution to this problem for Symbiodiniaceae. While perhaps unlikely, this issue could potentially explain the patterns seen, and should be acknowledged.

### Previous tests of incongruence

No previous study on Symbiodiniaceae seriously considers symbiont hybridisation, except that of *Wilkinson et al. (2015)*, which also finds evidence for its existence. However, aside from the potential examples of hybridisation mentioned in the Introduction of this study (*LaJeunesse et al., 2003*, *2004*; *LaJeunesse, 2005*), three other studies bear mention. *Sampayo, Dove & LaJeunesse (2009)* also focused on the basis that hybridisation can cause incongruence between genes from different organelles, and built trees from mitochondrial, chloroplast and rDNA nuclear gene regions to test this. Based on visual inspection of these trees, they concluded that different symbiont lineages (types) within *Cladocopium* are reproductively isolated. Interestingly, they did also use the ILD test to formally test incongruence, which returned a *p*-value of 0.01, though this result was not explored further. *Pochon, Putnam & Gates (2014)* assessed six genes from three different organelles (mitochondrion, nucleus and chloroplast). In all cases, they found evidence of incongruence between pairwise comparisons of genes, using the AU test. While they go on to discuss the implications for concatenation in some detail, the cause of these incongruences was likewise not explored further. Another study from *Pochon et al. (2006)* found the surprising result of incongruence between whole genera rendered from nr28S and cp23S data, using the SH test. However, when they removed all but two members of each clade, the test then showed congruence between datasets. This indicated incongruence was being caused by the accumulation of occasional within-clade mismatches between the nucleus and chloroplasts, something which is also broadly agreeable with a hypothesis of hybridisation in low frequency. These studies certainly do not provide conclusive evidence of hybridisation. However, it is reasonably striking that four studies conduct an explicit statistical test of incongruence within Symbiodiniaceae (*Pochon et al., 2006*, *Pochon, Putnam & Gates, 2014*; *Sampayo, Dove & LaJeunesse, 2009*; this study), and all four find evidence for its existence. At the very least, these add to the body of evidence that the family Symbiodiniaceae has not evolved in a simple linear fashion, and justifies a more careful consideration of patterns of incongruence within this family.

## CONCLUSIONS

This study cannot be considered unequivocal proof of *Cladocopium* hybridisation. However, the unambiguous evidence for incongruence between nuclear and organellar

gene regions shows the value of the stepwise approach taken here, and conforms to the hypothesis of hybridisation between divergent taxa. While ILS remains a possibility, it is a less intuitive explanation, especially in the light of incongruent samples having clearly distinct, predefined types which were recovered in non-incongruent samples, and the failures of background populations to consistently align to its predictions. Therefore, hybridisation appears to be a credible, if infrequent, mechanism for adaptive change in *Cladocopium*, and potentially for Symbiodiniaceae in general, though multiple sources of intragenomic variation remain analytically problematic. Ascertaining the frequency and extent of this may be vital to predicting the fate of coral reefs in an environmentally unpredictable future.

## ACKNOWLEDGEMENTS

Evan Raymond assisted with the optimisation of *actin* gene sequencing. We are grateful to two anonymous reviewers, whose comments improved the quality of this manuscript.

### Funding

This work was supported by a William Georgetti Scholarship awarded to Joshua I. Brian and a Rutherford Foundation Postdoctoral Research Fellowship to Shaun P. Wilkinson. The funders had no role in study design, data collection and analysis, decision to publish, or preparation of the manuscript.

### Grant Disclosure

The following grant information was disclosed by the authors:
A William Georgetti Scholarship to Joshua I. Brian and a Rutherford Foundation Postdoctoral Research Fellowship to Shaun P. Wilkinson.

### Competing Interests

The authors declare that they have no competing interests.

### Author Contributions

- Joshua I. Brian conceived and designed the experiments, performed the experiments, analyzed the data, prepared figures and/or tables, authored or reviewed drafts of the paper, approved the final draft.
- Simon K. Davy conceived and designed the experiments, contributed reagents/materials/ analysis tools, authored or reviewed drafts of the paper, approved the final draft.
- Shaun P. Wilkinson conceived and designed the experiments, performed the experiments, contributed reagents/materials/analysis tools, authored or reviewed drafts of the paper, approved the final draft.

### Field Study Permissions

The following information was supplied relating to field study approvals (i.e. approving body and any reference numbers):

The corals in this study were sampled with the permission of the Government of Timor-Leste (Ministerio da Agricultura e Pescas, permit number LNC-PC0012.VI.16).

## DNA Deposition

The following information was supplied regarding the deposition of DNA sequences:

The actin sequences are accessible at GenBank: MK520897–MK520906. This study also makes use of previously published sequences from GenBank: MH236749–MH236784 (cob, ITS2) and MH329431–MH329571 (psbA).

## Data Availability

Raw data files for the ILD and AU tests, in addition to annotated exemplar code to run both analyses, are available on GitHub: https://github.com/brianjosh/Cladocopium_alignments.

## Supplemental Information

Supplemental information for this article can be found online at http://dx.doi.org/10.7717/peerj.7178#supplemental-information.

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
