# Peer review of "Multi-gene incongruence consistent with hybridisation in Cladocopium (Symbiodiniaceae), an ecologically important genus of coral reef symbionts"

_PeerJ, doi:10.7717/peerj.7178_

## Round 0.1 · original submission · Minor Revisions

Two expert reviewers have evaluated your manuscript and their comments can be seen below and in a PDF. YOur manuscript is well-written on a difficult and understudied issue of possible hybridization in the Symbiodiniaceae.

One concern is that the methods section relies heavily on a recently published paper to explain the details of the study design. It would be a good idea to include more details in this methods section to ease flow..

Another concern is that the study is skewed by analysing only the most abundant sequences and what impact this may have on intragenomic ratios and their interpretation.

A third concern is about background symbiont populations.and the possible need for a figure to aid readers less familiar with this issue.

Due to these concerns I am recommending a revision, and encourage you to attend all of the concerns of the reviewers.

Reviewer 1 ·

Basic reporting

There were a few inconsistency of language/words, such as:
1) double negative (grammatically not encouraged)
2) there were some inconsistency of citation: "Wilkinson and colleagues" and "Wilkinson et al."

There are few suggestions on sorting some of the contents especially from DISCUSSION to either INTRODUCTION, METHODS or RESULTS.

The manuscript, especially in the METHOD section relied too much on referring the readers to one specific paper published previously. This left a lot of important details out in the manuscript, which caused some confusion.

Are the 27 alignments in this paper publicly available in repositories (i.e. GenBank?) or is it possible for the author(s) to submit the alignments as Supplementary data in nexus format?

(Please see attachment for more details)

Experimental design

The most important issue with the explanation of the study design is that the author relied on referring the readers to a specific paper previously published. This left a lot of details for the readers to find out themselves, which does not serve the purpose of independent study. Details are as below:

1) The study did not mention specifically what type/species hosts of the Cladocopium extracted from (L 143-160)
2)How many specimens in total does the author(s) collected? (L157-159)
3) What are the detailed procedures in amplifying the markers (cob, psbA and ITS2), with at least specify the markers used and methods modified from previous papers. (L161)
4) psbA is a hypervariate region, often impossible to make consensus between forward and reverse sequences, hence only one direction is used for alignments. Is it true for this study? (L166-167)
5) 'METHOD: Incongruence tests' section needs to be separated into more paragraphs for ease of reading and identify the different hypotheses tests. (L170-199)
6) The actin gene came in suddenly without explaining its contribution in this study in detail. Furthermore, the method ended abruptly at the sequencing phase. Perhaps the author can mention "...no further step taken and will be explained in the discussion"? (L288-301, L537)

(Please see attachment for more details)

Validity of the findings

There are a few inconsistency of language in explaining the data when it comes to the words "congruence", "incongruence", "no incongruence". These three are essential in explaining the data, explaining the data using only two out of the three will be much appreciated.

1) May I know what is the standard of ' approaching significance'? Providing citations is much appreciated. (L311-312)
2) The hypotheses should be established clearly in METHODS so readers will have an idea coming to the RESULTS. (L340-342)
3) LIW serves as the control for the post hoc test? (L349-353)
4) Thornhill et al. (2017) mentioned specifically that sexual recombination do happen but in a very low frequency (correct me if I am wrong). Do you think mentioning this will also prove that why most of your 'hybridized' Cladocopium present in a very low frequency? (L503, L570-572, L580-591)
5) The NGS results are important in explaining the hybridization section. However, this part is missing from the RESULTS section. Can you please write something in the results so readers can have the right idea going into the DISCUSSION. Initially, I was skeptical with the hybridization theory without this data in the results, as I was thinking there might be a co-dominant in play. Including the NGS in the RESULTS will be much appreciated. (L509-512)

(Please see attachment for more details)

Additional comments

Please see attachment for more details

Annotated reviews are not available for download in order to protect the identity of reviewers who chose to remain anonymous.

Reviewer 2 ·

Basic reporting

no comment

Experimental design

no comment

Validity of the findings

no comment

Additional comments

Brian et al. use Symbiodiniaceae sequencing data to look for incongruence in different gene regions, which would be consistent with (but not conclusive evidence of) hybridization among distinct species of coral symbionts. They apply several phylogenetic analyses to a set of 158 samples of symbiont communities residing in coral tissues, identifying 3 with clear patterns of incongruence between nuclear and organellar genomes in the most abundant symbiont sequences per gene. They discuss at length how the most parsimonious explanation is symbiont hybridization.

This is an extremely well written manuscript. The topic of Symbiodiniaceae hybridization is both understudied and controversial, and the authors deftly guide the reader through the nuances and difficulties of testing the hybridization hypothesis. They are quite forward about how their results are not definitive, but represent an exciting first step. Though I still have some “big picture” reservations about their methods, I find their execution sound and I appreciate their responsible approach to reporting their findings. Most of my suggestions are minor, though I do ask that the authors consider a potential problem that they don’t address in the manuscript (how the combination of sampling a symbiont community and using multicopy genes may lead to artificial incongruence signal when only the most abundant variants are selected for analysis).

My biggest concern crops up in L162-164. To arrive at one variant per gene in order to perform the incongruence analyses, the authors selected only the most abundant ITS2 variant within a given sample after amplicon sequencing. I have a general concern about combining this selective approach with Sanger sequencing of cob and psba. Here’s my line of thinking:

First, every sample of a given coral contains millions of individual symbiont cells. Second, the symbionts are a community of potentially several different species. Third, each symbiont species may encompass several genetically unique strains. Finally, the ITS2 region is part of a multicopy array, so within each unique symbiont genome, there is the potential for several sequence variants to exist for this gene. Although concerted evolution generally keeps sequence variants more similar within species than between species, the distribution of variants within a sample of a community can still be problematic. Certain variants may be maintained at different frequencies among different genomes within the same species or even across species. Sometimes a given variant is dominant in many different species, and other times, a single species contains different dominant variants depending on which individual cell is sampled.

So amplicon sequencing of ITS2 is going to capture a lot of variants from many different sources (interspecies, intraspecies, and intragenomic). However, direct Sanger sequencing of cob and psba is presumably only going to return the single most abundant variant (or whichever variant is most amenable to PCR amplification). While cob is fairly well conserved within a species and across individuals, psba, like ITS2, is multicopy. As far as I know, at this point it has not been established whether there is any association within a given genome between ITS2 variant abundance and psba variant abundance, and if so, to what extent these associations may vary in different populations.

Consider a hypothetical genome where ITS2 variant A represents 90% of the intragenomic variation, while ITS2 variant B represents 10%. In the same genome, psba variant 1 represents 90% of the intragenomic variation, while psba variant 2 represents 10%. We’ll call this the 9:1/9:1 genome to represent the ratios of variants for ITS2 and psba, respectively. It’s entirely possible that in another individual, the ratio is not the same; for example, you could get a 9:1/1:9 genome. Or a 9:1/4:6 genome. When millions of individuals with potentially different variant ratios across different genes are sampled from a symbiont community and the DNA is extracted, all of these variants are mixed into one giant pot.

Choosing the most abundant variant for a given multicopy gene (as was done for ITS2 after NGS, or through Sanger sequencing for psba) captures only the most abundant ratio after homogenizing all variants, but does not necessarily reflect the true gene sequence associations for the most abundant individuals within the population. I imagine in most cases it would, but I could also imagine rarer cases where a given population in a subset of samples happens to have a skewed intragenomic variant ratio distribution such that even though the most abundant sequence variants are represented WITHIN each gene region, the most abundant combination of variants ACROSS the three regions is not represented. I believe that this could manifest itself in the data as incongruence in a rare subset of samples, even though sequence combinations lacking incongruence could be found in isolation as distinct species entities in other community samples.

I think the only way to address this issue would be to sequence multiple markers from the same DNA strand through long-read sequencing, thus preserving the ratio of intragenomic variants within a given genome. Of course, that wouldn’t work for markers distributed across nuclear, mitochondrial, and chloroplast genomes. I don’t have a solution, but I do feel this issue needs to be acknowledged. The problem is unique to the way that the authors have sampled symbiont communities and then reconstructed gene region associations from only the most abundant sequences for multicopy genes. It’s a limitation of the technologies and genes we have available for Symbiodiniaceae phylogenetic analyses, and I feel it should be mentioned explicitly in addition to the other issues like independent lineage sorting.

My other major concern relates to the findings reported from L506-514 related to background symbiont populations. These results should be included in the Results section. They deserve a figure of their own as it’s hard to grasp the key concept without a visual. I’d recommend a background symbiont proportion diagram from the representative samples mentioned in the text.

L56: Should cite the genus revision here: LaJeunesse et al. 2018 Current Biology
L59: Change to “exclusively asexual in hospite”
L95: Change to “in several instances”
L127: Change to “warrants further study”
L131: Change to “genes are typically inherited biparentally”
L243-244: Why was the default changed?
L258-268: I didn’t quite follow how the reciprocal tests were evaluated to recognize whether a tree that was statistically significant was truly incongruent or not
L266: Please define “RELL bootstrap”
L444-446: You’re describing concerted evolution; I'd use the term here
L483: Change to “incongruence between nuclear”
L538: Delete “reasonably”
L541: Change to “lower-resolution than actin”

---

## Round 0.2 · accepted · Accept

I am satisfied with all of the modifications and additions that have been made to the manuscript.

#